# A Quantum Annealing Bat Algorithm for Node Localization in Wireless Sensor Networks

**DOI:** 10.3390/s23020782

**Published:** 2023-01-10

**Authors:** Shujie Yu, Jianping Zhu, Chunfeng Lv

**Affiliations:** College of Engineering Science and Technology, Shanghai Ocean University, No. 999, Huchenghuan Rd., Nanhui New City, Shanghai 201306, China

**Keywords:** wireless sensor networks, node localization, bat algorithm, geometric features, quantum evolution, tournament, natural selection

## Abstract

Node localization in two-dimensional (2D) and three-dimensional (3D) space for wireless sensor networks (WSNs) remains a hot research topic. To improve the localization accuracy and applicability, we first propose a quantum annealing bat algorithm (QABA) for node localization in WSNs. QABA incorporates quantum evolution and annealing strategy into the framework of the bat algorithm to improve local and global search capabilities, achieve search balance with the aid of tournament and natural selection, and finally converge to the best optimized value. Additionally, we use trilateral localization and geometric feature principles to design 2D (QABA-2D) and 3D (QABA-3D) node localization algorithms optimized with QABA, respectively. Simulation results show that, compared with other heuristic algorithms, the convergence speed and solution accuracy of QABA are greatly improved, with the highest average error of QABA-2D reduced by 90.35% and the lowest by 17.22%, and the highest average error of QABA-3D reduced by 75.26% and the lowest by 7.79%.

## 1. Introduction

WSNs are highly researched as an extension of the Internet to the physical environment in the Internet of Things [1]. WSNs are composed of a variety of small smart sensors that can collect, store, and transmit various environmental data according to different needs, facilitating more accurate monitoring of relevant dynamics [2].

In WSNs, only every node is accurately positioned, so that it can play a greater role [3]. Typically, only very few anchor nodes (ANs) in WSNs can be located by GPS or are artificially configured to determine their location, while the location of most unknown nodes (UNs) needs to be determined with the help of node location algorithms [4]. There are various algorithms used for node localization in WSNs, which are mainly classified into range-based localization algorithms and range-free localization algorithms based on whether ranging is required, among which range-based node localization algorithms include received signal strength indicator (RSSI) [5] and time of arrival (TOA) [6]. The range-free node-localization-based algorithms include distance vector hop (DV-Hop) [7] and centroid [8]. Each method has its advantages and disadvantages, and all problems cannot be solved by one fixed method. Therefore, comprehensive consideration of factors such as technical difficulty and environment is an important reference index for selecting appropriate methods.

Among the range-free node-localization-based algorithms, DV-Hop and Centroid algorithms are research hotspots all the time due to their simple structure and easy implementation. DV-Hop uses the number of hops between UNs and ANs and the average distance of each hop segment to estimate the location of UNs, which has the advantage of large localization coverage and less additional hardware [9]. Since ranging is not required, the positioning accuracy of such algorithms is easily affected by the jump range error. In order to improve the positioning accuracy, reference [9] changed the communication power of nodes and proposed a method of taking the average distance between an unknown node and three adjacent nodes as the reference value of hop distance. Weighted DV-Hop [10] proposed a method of adding the best weighting function to the hop number distance conversion formula, while the centroid algorithm uses the center of mass of an unknown node in a region to predict the position of the node. This method is simple to implement, but the localization accuracy is easily affected by the node density of the region. Therefore, the trilateral center-of-mass localization algorithm [11] used a RSSI-based weighting compensation scheme to eliminate ranging interference to change this situation. The 3D-weighted centroid localization algorithm (3D-WCL) [12], on the contrary, utilized the degree of influence of ANs on UNs to change the weighting factor, and then introduced the weighting factor into the centroid coordinates to reduce this influence.

In the ranging-based node localization algorithm, the localization accuracy is much higher than that of the ranging-free-based localization algorithm under the same conditions because the distance between UNs and ANs is accurately calculated. Therefore, RSSI-based and TOA-based localization algorithms are more widely used in the localization of WSNs, and the localization results are higher. Among them, RSSI-based positioning algorithms have received much attention from scholars because of their excellent performance with no additional hardware devices, low cost, low energy consumption, and easy implementation [13]. Moreover, they have been successfully applied in smart agriculture [14,15], power systems [16], smart buildings [17], military fields [18], and environmental monitoring [19,20,21,22].

Although RSSI-based positioning algorithms have structural and cost advantages under ideal conditions, their positioning accuracy are easily affected and significantly degraded by ranging errors in environments with high interference [23]. However, it is not only the anisotropic noise in the signal path that has an impact on the localization accuracy [24], but also the too small communication radius and sparse anchor nodes are the main causes. Therefore, to further improve the performance of node localization algorithms for WSNs, various heuristic algorithms with high adaptive capability and simple structure, as well as computational efficiency, are improved and are successfully used for node localization in WSNs [25].

In this paper, we propose a quantum annealing bat algorithm (QABA) for node localization in WSNs. The bat algorithm is used as a framework for QABA, and quantum evolutionary strategies and simulated annealing evolutionary mechanisms are combined to help improve the local and global search performance. Then, QABA achieves global and local search equilibrium with the help of tournament and natural selection convergence strategies, and continuously approaches the optimized value. Meanwhile, the RSSI-based 2D and 3D localization models and corresponding algorithms are developed respectively with the quantum annealing bat algorithm as the optimization core. Finally, the performance of the proposed algorithm is tested using 22 test functions and 2D and 3D WSNs node localization problems. The main contributions of this paper are as follows.

A quantum annealing bat algorithm that incorporates quantum evolutionary strategies and simulated degenerate evolutionary mechanisms is proposed as a framework for the bat algorithm.The convergent evolutionary strategies of tournament and natural selection are used to approach the optimum to balance the local and global search mechanisms, and QABA’s performance is verified with 22 basis test functions.A 2D localization model and a localization algorithm are developed, and the performance of the algorithm is verified by node localization problems.A 3D localization model and localization algorithm are developed, and the performance of the algorithm is verified by node localization problems.

The rest of this paper is distributed as follows: Section 2 describes the node localization algorithm for WSNs combined with the heuristic algorithm, as well as the source bat algorithm and its improvement strategy. Section 3 describes our proposed improved algorithm and the related pseudo-code. Section 4 simulates QABA in 22 test functions and 2D and 3D WSNs and analyzes the results. Section 5 summarizes the whole paper and gives future expectations.

## 2. Related Works

The node localization algorithm of WSNs combined with a heuristic algorithm has better localization performance in an interference environment, which is a more mainstream research direction at present. The heuristic algorithm is simple in structure and highly portable, and can be used for solving complex problems with multiple models. Especially in the node localization of WSNs, traditional localization algorithms are gradually being replaced by localization algorithms incorporating heuristic algorithms. Among them, the localization algorithms can be classified into 2D localization algorithms and 3D localization algorithms, according to the model structure. Due to the difference in spatial dimensions, they also differ in the difficulty of implementation.

The 2D localization algorithm needs to consider only two dimensions of the optimization search problem, and the fitness function and structure are simpler and more efficient than the 3D localization algorithm. In terms of specific algorithms, in the hybrid optimized localization algorithm (WOA-QT) [26], the co-evolutionary capability of the quasi-affine transform evolutionary algorithm (QUATRE) was fused with the whale optimization algorithm (WOA) to improve localization accuracy, but there was still the problem of falling into local optima. In the particle swarm optimization-based localization algorithm (IDE-NSL-AWSN) [23], ANs were grouped and cooperated with each other to increase the density of ANs, and then the particle swarm optimization algorithm (PSO) was used for localization to improve its localization accuracy and global search capability, but it still suffers from a lengthy structure and increased cost. While the Gaussian-modified RSSI algorithm and the improved whale optimization algorithm were used in the improved whale optimized localization algorithm (IWOA) [27] to reduce the ranging error and perform search localization, which improved the population diversity and node localization accuracy, there was still the problem of falling into local optimum. In addition, the hybrid gray wolf and firefly localization algorithm (GWO-FA) [28] used the concept of a fuzzy logic system to design a distance-free technique with an irregularity of 0.01 to improve the density of ANs, and it combined with GWO-FA to improve localization accuracy; yet, there is still room for improvement in optimizing the speed and accuracy. From the experimental results, the research focus of the above localization algorithm is mainly on how to improve localization accuracy, thus consuming too much energy. However, the firefly optimized localization algorithm (FA) [29], the K-value common line and gray wolf optimized localization algorithm (DCK-GWO) [30], the quantum optimized localization algorithm (QA) [31], and the anti-barrier hybrid localization algorithm (D-PSO and D-C) [32] have been studied in terms of energy consumption and are able to save more energy while obtaining higher accuracy.

The error tolerance of 3D localization algorithms is significantly better than that of 2D localization algorithms in real environments and operating conditions [33]. 3D localization algorithms are also classified as range-based [34] and range-free-based [35,36,37], but the models of localization algorithms are very different. The least squares method (LSM) [38] has advantages in algebraic operations, hence LSM is more used in 3D localization, but its susceptibility to singular matrices leads to the poor accuracy of the algorithm [39]. In contrast, among the 3D node localization algorithms using heuristics, the rotating black hole localization algorithm (RBH) [40], the large-scale robust localization algorithm (MHL-M) [41], and the motion target node localization algorithm [42] have higher adaptability, and their localization performance has been demonstrated in the respective literature. Although many algorithms show better localization performance in a single dimension, they are not validated in both 2D and 3D space at the same time and are not universally applicable.

The above algorithms have been improved by different evolutionary mechanisms, but the research on local search is ignored. Using appropriate local search methods in heuristic algorithms can improve the accuracy and success rate of the algorithms. The hybrid genetic algorithm (HMA) [43] combined with the Memtic algorithm with local search strategy to improve the local exploration capability of the algorithm and obtained higher accuracy in solving the node siting problem. The harmonic search-local search algorithm (HS-LS) [44] proposed a connectivity-based local search method and successfully combined with the Harmonic Search algorithm (HS) to alleviate the problem of sparse network deployment and reduced localization error in WSNs. This shows that the appropriate local search method is very important to improve the solution and positioning ability of the algorithm, and the balance between global search and local search will further strengthen this advantage, which will also guide the research direction of our work.

We introduce the advantages and disadvantages of combining a heuristic algorithm with WSNs from the perspective of improvement strategy and application. Besides the above algorithms, there are many heuristic algorithms with excellent performance waiting to be discovered. One of them is the bat algorithm (BA), with simple structure and high accuracy. BA is a new meta-heuristic algorithm proposed in 2010 by Yang [45], a scholar at the University of Cambridge, inspired by the predatory behavior of bats using acoustic echolocation. The solution success rate of BA is improved by continuously changing the pulse generation and loudness. When the loudness is large and the pulse generation rate is small, it is good for global search, while when the loudness is small and the pulse generation rate is large, it is good for local search. Similar to the same type of particle swarm algorithm (PSO) [46], genetic algorithm (GA) [47], differential evolution algorithm (DE) [48], sine cosine algorithm (SCA) [49], and other heuristics, BA still suffers from low convergence accuracy and the tendency to fall into local optimum when dealing with complex function problems with multiple constraints. Therefore, schemes such as improved operators [50], increased weights [51], and structural reorganization [52] are used by many scholars to improve the performance of BA, with desirable results. In addition, due to its few parameters and strong applicability [53], BA has been successfully used in image recognition [54], vehicle path planning [55,56,57], and wireless sensor networks [58]. The main structure model of the source bat algorithm is as follows.
(1)$$f_{i}=f_{\min}+\beta (f_{\max}-f_{\min})$$
(2)$$v_{i}^{t}=v_{i}^{t-1}+f_{i}(X_{i}^{t-1}-X_{*})$$
(3)$$X_{i}^{t}=X_{i}^{t-1}+v_{i}^{t}$$
where $f_{\max}$ and $f_{\min}$ denote the maximum and minimum values of frequency; β∈[0,1].

During the search process, as the loudness of its acoustic wave $A_{i}$ and the pulse generation rate $r_{i}$ are adjusted, the position equation of BA is updated as follows.
(4)$$X_{new}=X_{old}+\varepsilon A^{t}$$
where $X_{new}$ is the new bat position, $X_{old}$ is the global optimized individual position, ε∈[0,1], and $A^{t}$ is the average loudness of the population of the same generation.

The loudness $A_{i}$ and the pulse rate $r_{i}$ of BA are updated by the following equations.
(5)$$A_{i}^{t}=\alpha A_{i}^{t-1}$$
(6)$$r_{i}^{t}=r_{i}^{t-1}\left[1-\exp(-\gamma t)\right]$$
where the attenuation coefficient of loudness is α∈[0,1], and the enhancement coefficient of acoustic frequency is γ>0.

## 3. Our Proposed QABA Scheme

Our proposed quantum annealing bat algorithm mainly takes the bat algorithm as the framework, fuses the simulated annealing algorithm and quantum evolution strategy to improve the search capability, and keeps approaching the optimized solution under the guidance of the convergence mechanism of tournament and natural selection. In this section, we focus on the update mechanism and implementation steps of the improved algorithm, as well as the implementation steps of the two-dimensional node localization algorithm and three-dimensional node localization algorithm incorporating the improved algorithm.

### 3.1. Improved QABA

#### 3.1.1. Quantum Evolution Strategy

The quantum evolutionary strategy is an approach to quantum chromosome variation that has significant advantages in convergence and merit seeking of high-dimensional functions [59,60]. The evolutionary strategy of most standard algorithms involves unpredictable variation with the number of iterations, and such evolutionary variation methods often require large population sizes and high population diversity. However, the direction of quantum evolutionary variation can be guided by using the current optimized and local optimized values to distribute the subgenerational population around the optimized individuals, thus allowing the convergence speed and the accuracy of the algorithm to be effectively improved. The quantum evolution equation is as follows.
(7)$$x_{i}^{t}=(P\pm \lambda \times \left|Q_{mean}-x_{i}^{t-1}\right|\times \ln\frac{1}{\delta })\times f_{i}{}^{\prime }$$
(8)$${f^{\prime }}_{i}=f_{i}\times \frac{1}{t^{2}+1}$$
(9)$$P=\mu_{1}\times Q_{best}+(1-\mu_{2})\times G_{best}$$
(10)$$\lambda =\frac{\left(1-0.5)\times (T-t\right)}{T}+0.5$$
(11)$$Q_{mean}=\frac{1}{N}\sum_{i=1}^{N}Q_{i}$$
where P is the guiding quantum chromosome; λ is the scaling factor; $Q_{mean}$ is the mean value of the current population position; $Q_{best}$ and $G_{best}$ are the local optimal individual position and the global optimal individual position, respectively; T is the maximum number of iterations; and $\mu_{1},\mu_{2},\delta \in \left[0,1\right]$, N is the population size.

#### 3.1.2. Metropolis Sampling Guidelines

The Metropolis sampling criterion is a key part of the Simulated Annealing Algorithm (SAA) that helps the SAA to direct the population that wanders randomly in space toward the optimal value, thus improving the solution efficiency of the SAA [61,62]. The principle of the criterion is that when the optimal solution is received, the poor solution can also be accepted with a certain probability, so as to increase the diversity of solutions and improve the solution efficiency.

#### 3.1.3. Assisted Convergence Evolutionary Strategy

Although the above evolutionary strategy has a greater advantage in population convergence evolution, it is deficient in the balance between local and global search optimization after population mutation evolution. In order to make the algorithm more coherent in the overall search for optimization and improve the solution efficiency, the population after mutation is assisted to converge. In terms of local and global search balance, a dynamic weight that can dynamically adjust the search range with the iteration and annealing process is added to the speed update formula of BA, which will effectively improve the convergence performance of the algorithm. Larger inertia weights can enhance the global search capability, and smaller inertia weights can enhance the local search capability of the algorithm [63]. The dynamic weight and speed update formula is as follows.
(12)$$\omega =1-n\frac{i}{T}\times \frac{t}{T}$$
(13)$$v_{i}^{t}=\omega v_{i}^{t-1}+f_{i}(X_{i}^{t-1}-X_{*})$$
where n is a constant, i is the current number of bats, t is the current number of iterations, T is the total number of iterations, ω is the adaptive dynamic weight factor, and $X_{*}$ is the global optimal bat position.

In the assisted convergent evolution scheme, improved tournament and natural selection mechanisms are used to improve the ability to jump out of the local optimum. The traditional tournament selection is to randomly select a number of offspring individuals from the current optimized local position each time, then rank the fitness values of the selected offspring individuals and take the best value of the fitness value into the offspring population. We improve it by randomly selecting several elements of child individuals each time, with the population size as the boundary and the first column of the population as the target, and then recombining these elements into several new individuals. The specific principle is shown in Figure 1.

The principle of the natural selection strategy is that during an iterative update, the resulting current locally optimized solution is ranked and divided equally into two. The better half of the solution is retained, while the position of the worse half is replaced with the position of the better half, thus forming a new population. An improved solution is to set up a comparison mechanism, where a smaller probability p is introduced to randomly accept the worse solution when selecting the offspring population. In this way, the diversity of the population is better preserved, and the effect of the worse solution on the convergence speed and accuracy of the algorithm is eliminated, and the local performance of the algorithm in finding the best is improved. Finally, the pseudo-code of QABA is shown in Algorithm 1.
**Algorithm 1:** Pseudo-code for the QABA algorithmInput: A, r, f, dimensionality D, α, γ, population size N, gen, initial temperature T, temperature decay coefficient α1. Output: Global optimal position bestX, global optimal fitness value bestY.   1: Distribute the population X, calculate the fitness value Y, and determine bestX and bestY.   2: for t = 1:gen   3: Update X using tournaments and natural selection, and update bestX and bestY.   4: for i = N   5:  Update the population according to Equations (1), (3), (12) and (13) and determine the local optimal position pbesX0 and the fitness value pbestY0.   6:  Update X with a quantum evolutionary strategy and determine pbestX1 and pbestY1.   7:  If pbestY0 < pbestY1   8:   Update pbestX and pbestY by taking the smaller one.   9:  end if   10:   Update pbestX and pbestY with the Metropolis criterion.   11:   if pbestX <= Y(i) && rand (0,1) < A(i)   12:    Update X(i), Y(i), A(i) and r(i).   13:    if pbestY < bestY   14:     Update bestX and bestY.   15:    end if   16:  end if   17:  end for   18: Perform annealing operation.   19: end for

### 3.2. Two-Dimensional Spatial Node Localization Algorithm

In RSSI-based localization algorithms, the position of UNs can be estimated from the received signal strength [64]. The trilateral localization algorithm is a commonly used node localization algorithm, which implements the basic principle of localization as follows: three ANs are located as $P_{1}(x_{1},y_{1})$, $P_{2}(x_{2},y_{2})$, and $P_{3}(x_{3},y_{3})$ if there exists an unknown node located as Q(x,y) and their distances are known as $d_{1}$, $d_{2}$, and $d_{3}$. Then, the coordinate positions of the three ANs and the corresponding communication radius are used as the center and radius to draw three circles respectively, and their only intersection point is the unknown node position. The specific principle is shown in Figure 2.

From the principle of the classical trilateral localization algorithm, it is known that at least three ANs are needed to achieve the localization of UNs. When the ANs are closer to the UNs, the more they are and the larger the radius, the relatively higher the positioning accuracy is.

A suitable and accurate fitness function can guide the search direction of the BA and improve the efficiency of the solution [65]. The fitness function of this model is formulated as follows:(14)$$F(x,y)=\frac{1}{m}\sum_{i=1}^{m}(\sqrt{{\left(x-x_{i}\right)}^{2}+{\left(y-y_{i}\right)}^{2}}-{\hat{d}}_{i}{)}^{2}$$
where m is the number of ANs involved in localization and m≥3. (x,y) is the location of UNs, $\left(x_{i},y_{i}\right)$ is the location of the *i*th anchor node, and ${\hat{d}}_{i}$ is the distance estimate of ANs and UNs.

The error between the UNs position and the predicted position is an important indicator of the localization performance of the algorithm. To ensure the authority of the overall localization error, we take the mean value of k errors as the final error. The error equation of the model follows.
(15)$$Err=\frac{1}{k}\sum_{i=1}^{k}sqrt({\left(x_{i}-{\hat{x}}_{i}\right)}^{2}+{\left(y_{i}-{\hat{y}}_{i}\right)}^{2})$$
where k is the number of UNs, $\left(x_{i},y_{i}\right)$ is the actual location of the *i*th unknown node, and $\left({\hat{x}}_{i},{\hat{y}}_{i}\right)$ is the estimated location of the *i*th unknown node.

To improve the distance measurement accuracy between UNs and ANs, the average value of *M* measurements after removing the extreme values is taken as the reference value of the distance. However, due to the limited energy of the sensor nodes themselves, energy will be excessively consumed if each node transmits too much data continuously. The following experiments were done to determine the value of M each time, while satisfying the accuracy. The algorithm parameters were set as follows: noise variance was 0.5, the number of ANs was 18, the number of UNs was 100, and the node radius was 30. The average of 30 consecutive runs of the algorithm at each M value was taken as the error reference value. The values are taken as shown in Figure 3.

According to the localization error curves of the different number of measurements in Figure 3, we can see that the optimal localization error of the algorithm has been approached at the number of 150; thus, we choose the number of 150 for each measurement. In addition, there is the problem that the localization error increases when the number of measurements increases in Figure 3. This is because the presence of too few anchor nodes near the unknown nodes increases the localization error, and the noise we add in each experiment also increases the localization error, while the same phenomenon can occur if our QABA runs unstably.

The 2D spatial localization algorithm is more widely used and has the advantages of easy implementation in the local area and high localization accuracy in practical applications. According to the 2D spatial localization model, we give the pseudo-code of the 2D spatial node localization algorithm with QABA as the core operation. Pseudo-code of QABA-2D is shown in Algorithm 2.
**Algorithm 2:** Pseudo-code for the QABA-2D algorithmInput: Communication radius R, AN, UN, noise variance VR, number of range repetitions PN, other parameters in the QABA algorithm.
Output: Minimum mean error besterr.   1: Random distribution of the locations of ANs and UNs.   2: for i = 1:UN   3:  Add noise to the distance and measure PN times, and take the mean value of distance d1 × AN after removing the extreme values.   4:  Arrange d1 × AN and determine the number BN of ANs less than R.   5:  if BN ≥ 3   6:  Select all ANs that satisfy the condition, otherwise select the three ANs closest to the UNs.   7:  end if   8:  Use QABA to calculate Equations (14) and (15), and obtain the minimum error.   9: end for

### 3.3. Three-Dimensional Spatial Node Localization Algorithm

The principle of node localization in 3D space can be described as follows: a known unknown node O(x,y,z) and any three non-coincident ANs $A(x_{1},y_{1},z_{1})$, $B(x_{2},y_{2},z_{2})$, and $C(x_{3},y_{3},z_{3})$ in space can form a spatial tetrahedron with six sides of lengths a, b, c, d, e, f. Figure 4 shows the spatial position of UNs in relation to ANs, where h is the height of the spatial tetrahedron and $O^{\prime }(x^{\prime },y^{\prime },z^{\prime })$ is the vertical foot of the height of the tetrahedron.

First, the area S of the base of the tetrahedron is found using Heron’s formula, as follows.
(16)$$S=\sqrt{p\left(p-d\right)\left(\left(p-e\right)\right)\left(\left(p-f\right)\right)}$$
(17)$$p=\frac{d+e+f}{2}$$
where S is the area of the tetrahedron base ∆abc; d, e, f are the three sides of the base; and P is a component of Herren’s formula.

The volume V of the tetrahedron is then solved using Euler’s tetrahedron formula, as follows.
(18)$$V^{2}=\frac{1}{36}\left|\begin{array}{ccc} a^{2} & D_{a} & D_{b}\\ D_{a} & b^{2} & D_{c}\\ D_{b} & D_{c} & c^{2} \end{array}\right|$$
(19)$$D_{a}=\frac{a^{2}+b^{2}-f^{2}}{2}$$
(20)$$D_{b}=\frac{a^{2}+c^{2}-e^{2}}{2}$$
(21)$$D_{c}=\frac{b^{2}+c^{2}-d^{2}}{2}$$
where V $V^{2}$ is the square of the volume of the tetrahedron derived from Euler’s tetrahedron formula; $D_{a}$, $D_{b}$, and $D_{c}$ are the components of Euler’s tetrahedron formula; and the letters a, b, c, d, e, f are the six side lengths of the tetrahedron.

Next, the height h of the tetrahedron can be found from the formula for the volume of the tetrahedron, as follows.
(22)$$h=\frac{V}{S}\times 3$$
where h is the height of the tetrahedron, V is the volume of the tetrahedron, and S is the base area of the tetrahedron.

The coordinates of the vertical projection $O^{\prime }$ point of the unknown node above the plane H can be found using the trilateral localization algorithm. The position of the point $O^{\prime }$ on the plane H is shown in Figure 5, where $L=\left\{l_{1},l_{2},l_{3}\right\}$ is the distance from the $O^{\prime }$ point to the three ANs.

We propose the fitness function as follows.
(23)$$f(x,y,z)=\sum_{i=1}^{n}\frac{1}{n}(\sqrt{{\left(x-x_{i}\right)}^{2}+{\left(y-y_{i}\right)}^{2}+{\left(z-z_{i}-h\right)}^{2}}-L_{i}{)}^{2}$$
where n is the number of ANs involved in localization.

The z-value of UNs is susceptible to accuracy deviations due to the height h of the spatial tetrahedron. Therefore, from the principle shown in Figure 3, the height h of the tetrahedron is updated using the Pythagorean theorem, as follows.
(24)$$\hat{h}=\frac{1}{n}\sum_{i=1}^{n}\sqrt{R_{i}{}^{2}-L_{i}{}^{2}}$$
where R={a,b,c} is the distance from UNs to ANs and $\hat{h}$ is the average value of h.

Finally, the coordinates of UNs are updated as follows.
(25)$$x=\left(x,y,z^{\prime }+\hat{h}\right)$$
where $z^{\prime }$ is the value of the z-axis of the vertical projection of UNs on the plane H.

The error formula we propose to measure the performance of the algorithm is as follows.
(26)$$error=\sqrt{{\left(x^{*}-x\right)}^{2}+{\left(y^{*}-y\right)}^{2}+{\left(z^{*}-z\right)}^{2}}$$
where $\left(x^{*},y^{*},z^{*}\right)$ is the real coordinate of Uns.

The 3D positioning algorithm is closer to the real situation and has a higher degree of confidence in the positioning effect. We give the pseudo-code of the 3D spatial node localization algorithm using QABA as the core operation according to the 3D spatial localization algorithm model. Pseudo-code of QABA-3D is shown in Algorithm 3.
**Algorithm 3:** Pseudo-code for the QABA-3D algorithmInput: R, AN, UN, noise variance VR, other parameters in QABA.
Output: Minimum mean error besterr.   1: Random distribution of the locations of ANs and UNs.   2: for i = 1:UN   3:  while S_abc_ == 0    4:  Calculate the distance dua between UNs and all ANs, and select the three smallest ANs.   5:  Calculate the area of the triangle Sabc according to Equations (16) and (17).   6:  Calculate the volume Vabc of the space tetrahedron according to Equations (18)–(21).   7:  Calculate the height dh of the tetrahedron according to Equation (22).   8:  If the three anchor nodes are not co-linear, jump out of the loop.   9:  end while   10:  Add noise interference to the high dh of the tetrahedron.   11:  Use the Pythagorean theorem to find the distance from the vertical foot O’ to the three ANs.   12:  Use QABA to solve Equations (23) and (26).   13:  Correct the position of UNs with Equations (24) and (25).   14:  Update the positioning error again.   15: end for

## 4. Simulation Results and Analysis

### 4.1. Test Functions Simulation Results and Analysis

We selected 22 standard Benchmark test functions [66,67] to verify the computational performance of the QABA algorithm. In order to make the selected test functions representative to fully test the performance of QABA, these functions have the following characteristics. Functions F1–10 are unimodal functions that can fully test the ability of QABA to converge to the global optimum; Functions F11–18 are multimodal functions that can verify the ability of QABA to jump out of the local optimum; Functions F19–22 are fixed dimensional functions that can test the ability of QABA to solve for a specific value in a fixed dimension. The dimensionality and optimal values of the functions are shown in Table 1, and information such as the specific formula of the functions can be found in the literature [66,67].

To further demonstrate the superiority of QABA, its operational results were compared with the Improved Quantum Annealing Bat Algorithm (IQBA) [68], Group Evolution Hybrid Bat Algorithm (LMBA) [69], Bat Differential Hybrid Algorithm (BADE) [70], Adaptive Weighted Mean Particle Swarm Algorithm (MAWPSO) [71], Differential Evolution Algorithm (DE) [48], Sine Cosine Algorithm (SCA) [49], and Bat Algorithm (BA) [45], under the same conditions.

Each function is run 30 times consecutively in the experiment, and the optimized value (OV), average value (AV), and standard deviation (SD) are selected to verify the algorithm performance. The population size of QABA is 30, the maximum number of iterations is 200, the initial temperature T0 = 3000, the loudness is 0.9, the frequency range is [−1,1], the temperature decay coefficient and the decay coefficient of loudness are 0.9, and the enhancement coefficient of pulse occurrence is 0.6. The environment in which the algorithm runs is Matlab R2016b. The population size and maximum number of iterations of the other algorithms are the same as those of this paper, and other parameters are shown in the respective original papers. The symbols ‘+’, ‘−’, and ‘=‘ in Table 2 and Table 3 indicate that the other algorithms have ‘better results’, ‘worse results’, or ‘similar results’ than/as QABA, respectively.

As shown in Table 2, QABA converged to near the minimum for all 22 tested functions, and 12 functions converged to the theoretical optimum. In addition, 13 functions had better results compared to IQBA and LMBA, and 9 functions had similar results. Compared with BADE, 14 functions had better results, and 8 functions had similar results. Compared with MAWPSO, there were 12 functions with better results and 10 functions with similar results. Compared with DE and BA, there were 15 functions with better results and 7 functions with similar results. Compared with SCA, there were 16 functions with better results and 6 functions with similar results.

In Table 3, the mean and standard deviation of 8 algorithms in 22 functions show that our proposed QABA had a strong advantage in terms of finding accuracy. Among them, QABA converged to the theoretical optimum in the mean value of 5 out of 10 unimodal functions. Eight functions performed better than IQBA, LMBA, BADE, MAWPSO, DE, SCA, and BA in the test; just two functions performed similarly to DE in the search for the optimized value; and one function performed similarly to the other six algorithms.

Among the eight multimodal functions, QABA converged to the theoretical optimum in two functions. Among them, four functions had similar solving performance and four functions had better find ability compared to IQBA; three functions had similar solving ability and five functions had better finding ability compared to MAWPSO; and only seven functions had better solve performance and one function had similar solving ability compared to LMBA, BADE, DE, SCA, and BA.

Among the four fixed-dimensional functions, QABA had better performance in finding the optimized performance for two functions and similar solving ability for two functions compared with IQBA. Compared with LMBA and BADE, both had higher computing performance for only one function and similar computing ability for three functions. Compared with the other five algorithms, the solving performance was better for all four functions.

The above results demonstrate that the algorithm structure of QABA, with BA as the framework and incorporating quantum evolution and annealing convergence strategies, has a great advantage in the optimization of high-dimensional functions. It also shows that the quantum annealing strategy can substantially improve the algorithm’s optimization-seeking accuracy and population diversity, and the tournament- and natural selection-assisted convergence strategies can also effectively regulate the local and global search balance problems. However, the population diversity variation mechanism of QABA still needs to be improved when solving negative value functions, especially fixed dimensional functions with large peak variations.

To better demonstrate the results of the above analysis, the convergence curves of the 8 algorithms in 22 functions are plotted in Figure 6, Figure 7 and Figure 8. Due to the high convergence accuracy (10^−100^ or more) of our proposed QABA and other algorithms, the conventional linear curve plots can no longer distinguish the convergence speed and accuracy of the eight algorithms. We use the linear type (y = k x, where the slope k is negative) in the 22 functions in Figure 6, Figure 7 and Figure 8 only in functions f10, f18, f20-22. When the curve is steeper, it means that the smaller the value of y corresponding to the X-axis in the curve, the better the accuracy and convergence speed of the algorithm. The Y-axis coordinates of the remaining 17 functions are presented in logarithmic form (y = loga^x^), and the Y-axis still varies linearly with the X-axis.

As can be seen in Figure 6, the fitness values in Figure 6a decrease as the number of iterations increases. Among them, QABA has converged to 10^−80^ at the number of iterations close to 25, while the other algorithms only converge to around 10^−10^. Moreover, it is known from the nature of the logarithmic function that its independent variable (natural numbers) takes values in the range of (0,+∞), where the bottom number of the logarithmic function is 10 and the natural numbers are the fitness values. Since the natural numbers cannot be taken to 0 in the logarithmic functions, QABA stops changing with the number of iterations after converging to 0. This also explains why QABA stops changing when it iterates around 25 in Figure 6a, while other functions can be shown in full. Similarly, in Figure 6b–f, there are cases where QABA is already much more accurate than the other algorithms when the number of iterations is less than 200. Moreover, in Figure 6a–f, the fitness value of QABA is always the smallest when the number of iterations is the same, thus further indicating that the convergence speed and solution accuracy of QABA are significantly better than other algorithms.

As can be seen from Figure 7, QABA has converged to 0 in Figure 7c,e,g,h when the number of iterations is less than 50, and it always has the smallest fitness value in the other functions, as well, indicating that QABA has a higher convergence speed and accuracy than the other algorithms.

It can be seen from Figure 8 that the fitness values of QABA and the other algorithms decrease with the increase of the number of iterations. Among them, QABA has a worse average accuracy than DE, except in function Figure 8d. Yet, in other functions, its fitness value is always no greater than other algorithms in the same number of iterations, thus indicating that QABA has a higher convergence speed and accuracy.

### 4.2. Simulation Analysis of 2D Spatial Positioning in WSNs

In the previous chapter, the advantages of QABA in terms of finding accuracy and convergence speed were demonstrated by the operation of standard functions. To further verify the localization performance of QABA in real 2D space, the actual localization results of the QABA and BA algorithms were first compared. Then, the localization errors of QABA-2D were compared with IQBA-2D, LMBA-2D, BADE-2D, MAWPSO-2D, DE-2D, SCA-2D, and BA-2D after different communication radii, a different number of ANs, and adding different noise variances. The 8 algorithms of the 2D localization algorithm were simulated in a 100 m × 100 m area with the number of UNs 100. Other variables were shown specifically in the experiments, and other parameters of the source algorithm remain unchanged.

#### 4.2.1. Positioning Effect and Analysis of QABA and BA

To demonstrate the improved performance of QABA over BA, the actual localization effect of both was plotted in Figure 9. In the simulation, the number of ANs for both was 30, the number of UNs was 100, and the node communication radius was 30.

From Figure 9, it can be seen that all 100 UNs were pinpointed successfully in QABA-2D, while only some nodes were localized in BA-2D, and the localization accuracy was poor. Thus, it can be proved that the QABA-2D algorithm improves the solution performance of the algorithm by improving the search structure, and it has higher localization accuracy than BA-2D in node localization in 2D space.

#### 4.2.2. Effect of Communication Radius on Positioning Accuracy

The communication radius affects localization accuracy mainly by influencing whether the algorithm can select a suitable anchor node for localization [72]. To explore a suitable communication radius, we set the communication radius interval of the anchor nodes to 10–50 m in our experiments. A total of 8 localization algorithms were used to localize 100 UNs under the interference condition, with the number of ANs 18 and the noise variance 0.5. Finally, the mean values of 30 consecutive runs under each communication radius range were taken to compare the performance of the 8 localization algorithms.

As can be seen in Figure 10, the positioning accuracy of the eight localization algorithms showed an overall trend of improvement with the increase of the communication radius. The reason for the accuracy improvement is that the distance between ANs and UNs is shortened by increasing the node communication radius, and the number of nodes involved in localization is increased, which makes ranging more accurate. In addition, the increase of the communication radius allows anchor nodes that fit the model better to be found, which can also improve the positioning accuracy.

Among them, from the trend of the error curve, QABA-2D had the best positioning performance and the highest accuracy, IQBA-2D had the worst positioning performance and the lowest accuracy, and the positioning performance of the other six positioning algorithms was closer to that of QABA-2D. From the average of 9 positioning errors, the average error of QABA-2D was 1.21 m, which was 90.35%, 71.06%, 62.16%, 63.87%, 47.91%, 25.83%, and 71.43% lower than the average error of the other 7 algorithms. Thus, it is proved that the QABA algorithm is superior to the other algorithms in finding the best performance, and it can save more energy and improve the working time for sensor nodes.

#### 4.2.3. Effect of ANs Number on Positioning Accuracy

A suitable number of anchor nodes can help the algorithm select more favorable anchor nodes for localization, thus improving localization accuracy and reducing energy consumption. In order to explore this issue, we set the variation interval of the number of anchor nodes to 9–30 in our experiments. A total of 8 localization algorithms were used to localize 100 UNs under interference conditions, with nodes’ communication radius of 30 and noise variance of 0.5. Finally, the mean values of 30 consecutive runs at each number of ANs were taken to compare the performance of the 8 localization algorithms.

From Figure 11, it can be obtained that the positioning accuracy of the eight positioning algorithms shows an overall trend of improvement with the increase of the communication radius. The increase in accuracy is due to the increase in the number of ANs, which indirectly shortens the distance between ANs and UNs, and the interference of other factors on the ranging accuracy is reduced. In addition, the increase in the number of anchor nodes likewise enables the algorithm to select more suitable anchor nodes for localization and achieve reduced localization errors.

Among them, from the trend of the error curve, QABA-2D had the best positioning performance and the highest accuracy, IQBA-2D had the worst positioning performance and the lowest accuracy, and the positioning performance of the other six positioning algorithms was closer to QABA-2D. The average error of QABA-2D was 1.25 m, which was 89.55%, 69.51%, 61.66%, 60.19%, 46.58%, 17.22%, and 70.59% lower than the average error of the other 7 algorithms. Thus, it is demonstrated that the improved structure and assisted convergence strategy of QABA improves the population diversity and overall search performance compared with other algorithms, and it is able to obtain higher localization accuracy with a smaller number of ANs and improve the lifetime of working nodes.

#### 4.2.4. The Effect of Different Levels of Interference on Positioning Accuracy

Different levels of noise cause different effects on the localization algorithms. Since the experiments are performed in a simulation environment, when we consider adding a certain noise with a signal-to-noise ratio value of SNR to a known signal, it is added not in the form of db, but in the form of heteroskedasticity. The study in the literature [24] also proved that the magnitude of the noise variance is proportional to the distance; hence, we set the range of the noise variance to 0.1–1 to verify the effect of noise on localization accuracy under different path loss. A total of 8 localization algorithms were used to localize 100 UNs, with a node communication radius of 30 and a number of ANs of 18. Finally, the mean values of 30 consecutive runs under each noise interference were taken to compare the performance of the 8 localization algorithms.

As can be seen in Figure 12, the positioning accuracy of the eight positioning algorithms decreases as the noise variance increases. The reason for the decrease in accuracy is that the noise interference becomes larger, and the accuracy of ranging between ANs and UNs is reduced.

Among them, from the trend of the error curve, QABA-2D had the best positioning performance and was least affected by interference, IQBA-2D had the worst positioning performance and the lowest accuracy and the positioning performance of the other six positioning algorithms was closer to QABA-2D. The average error of QABA-2D was 1.28 m, which is 89.19%, 69.74%, 62.57%, 60.00%, 45.53%, 20.50%, and 71.74% lower than the average error of the other 7 algorithms. Thus, it is proved that the QABA-2D algorithm has the advantage of superior seeking performance and distance processing method, which can reduce the influence of interference on positioning accuracy.

In addition, it can be seen from the different algorithms’ localization error variations that QABA using the quantum evolution strategy proposed in this paper has excellent performance in localization accuracy, while the IQBA algorithm has larger errors in localization. This is because the frequency of the guiding quantum chromosome of the quantum evolution strategy in this paper is quantized by BA, which enables the algorithm to obtain higher accuracy in high-dimensional optimization with the same population of individuals. However, the population being quantized will lead to the reduction of the population diversity of the algorithm, and it is easy to fall into local optimum, thus the solution accuracy of IQBA is poor. The tournament-assisted mechanism of QABA and BA cooperation can guarantee that the population diversity of the algorithm is not destroyed, thus making QABA more accurate and applicable in different problems.

### 4.3. Simulation Analysis of 3D Spatial Positioning in WSNs

In the previous section, it was demonstrated using real simulation data that the various improved strategies of QABA still have excellent performance in 2D spatial localization. However, 3D spatial localization increases the complexity of the model, the dimensionality of the fitness function becomes larger, and the variety of combinations of UNs locations increases substantially compared to 2D spatial localization.

All these changes put the localization performance of QABA-3D to the test. To prove that QABA still has advantages in 3D spatial localization, QABA-3D is simulated and compared with IQBA-3D, LMBA-3D, BADE-3D, MAWPSO-3D, DE-3D, SCA-3D, and BA-3D for localization errors, under the same conditions. Where ANs were randomly distributed in a stereoscopic area of 100 m × 100 m × 100 md, the number of UNs was 100, the communication radius was 50, and other variables were shown in specific experiments.

#### 4.3.1. QABA-3D and BA-3D Positioning Effect and Analysis

In the experiment, the number of ANs was 30, the number of UNs was 100, the node communication radius was 50, and all ANs were randomly placed at 1m from the ground. The actual positioning effect is shown in Figure 13.

From Figure 13, the positions of 100 UNs were successfully located by QABA-3D and show excellent localization accuracy. In contrast, only a very small number of nodes were precisely located by BA-3D, and the overall localization accuracy was poor. The experiment illustrates that QABA-3D still has better localization performance and higher accuracy than the source BA-3D in 3D node localization, thus further demonstrating the effectiveness of QABA’s auxiliary strategies, such as improved structure and tournament.

#### 4.3.2. Effect of Different Number of ANs on Positioning Accuracy

An appropriate amount of anchor nodes can help the algorithm select better anchor nodes for localization, thus improving the localization accuracy. In 3D localization, if the anchor nodes are selected closer to the unknown nodes and they form a more regular spatial tetrahedron, the localization error will be smaller. To investigate the influence of the number of anchor nodes on the localization accuracy, we set the distribution of the number of anchor nodes in the range of 5–30 and the noise variance at 0.5 in our experiments, and 100 unknown nodes were localized. Among them, each anchor node number variable was run 30 times individually, and the average value of the localization errors of 30 times was taken to compare the localization performance of QBAB-3D with other localization algorithms. The localization errors are shown in Figure 14.

As can be seen in Figure 14, the overall localization accuracy of the different algorithms increases with the increase in the number of ANs. The accuracy improvement is due to the increase in the number of ANs, and the distance between UNs and ANs is shortened, which leads to the reduction of the ranging error. Meanwhile, the increase in the number of anchor nodes will enable the algorithm to select more suitable anchor nodes, making the spatial tetrahedral structure composed of unknown and anchor nodes more regular, thus improving the search efficiency.

From the error curve, the localization accuracy of QABA-3D was always the highest; the localization performance of BADE-3D, MAWPSO-3D, DE-3D, and SCA-3D was similar to that of QABA-3D; and the localization performance of IQBA-3D, LMBA-3D, and BA-3D was poor. In terms of the average error, the localization error of QABA-3D was 2.25 m, which is 70.16%, 74.32%, 40.48%, 36.97%, 7.79%, 27.65%, and 67.53% lower than other localization algorithms, respectively. The results illustrate that the improved strategy of QABA is still applicable in 3D spatial localization to obtain higher accuracy and reduce node energy consumption with fewer ANs.

#### 4.3.3. The Effect of Different Levels of Noise on Positioning Accuracy

Similar to 2D localization, we set the range of noise to 0.1–1 in our experiments to investigate the effect of noise on localization accuracy under different path loss. The number of anchor nodes was 15, and the number of unknown nodes was 100. Each noise variable was run 30 times continuously, and the average of 30 times was taken to predict the performance of QABA-3D with other localization algorithms. The localization error curves are shown in Figure 15.

As can be seen in Figure 15, the overall localization accuracy of the different algorithms decreases as the noise variance increases. The decrease in accuracy is due to the increase in noise variance, which decreases the ranging error between UNs and ANs, which in turn affects the accuracy of the high spatial tetrahedra involved in localization.

In terms of error curves, QABA-3D always had the highest localization accuracy; BADE-3D, MAWPSO-3D, DE-3D, and SCA-3D had similar localization performance to QABA-3D; and IQBA-3D, LMBA-3D, and BA-3D had poor localization performance. In terms of the average error, the localization error of QABA-3D was 2.14 m, which was 72.21%, 75.26%, 37.06%, 42.01%, 9.32%, 32.92%, and 68.71% lower than other localization algorithms, respectively. The results demonstrate that QABA-3D outperforms similar algorithms in terms of search strategy, and it has high accuracy and anti-interference performance in localization.

## 5. Conclusions

In this paper, 2D and 3D localization algorithms are designed for the localization of nodes in different dimensions of wireless sensor networks using trilateral localization and geometric feature principles, and a quantum annealing bat algorithm (QABA) is proposed to improve the localization accuracy and applicability. Quantum evolution and simulated annealing convergence evolution strategies are incorporated into QABA, with the bat algorithm as a framework to improve the overall operational structure. Meanwhile, to balance the overall search and convergence problems of the algorithm, tournament and natural selection mechanisms are introduced to assist in fast convergence optimization. In the simulation of 22 standard functions, QABA outperformed the comparison algorithm on average for 14 functions, and was close to the other algorithms for 8 functions.

In addition, the QABA-2D localization algorithm and QABA-3D localization algorithm, which use QABA as the operation core, also show excellent localization accuracy and applicability in 2D and 3D spatial localization. The simulation results show that the average positioning error of QABA-2D in 2D space positioning was reduced by 17.22–90.35% compared with other algorithms, and the average positioning error of QABA-3D in 3D space was reduced by 7.79–75.26% compared with other algorithms. Thus, the results show that the proposed QABA not only has excellent performance in the standard function test, but also has excellent solution accuracy and applicability in node localization optimization of wireless sensor networks.

Future efforts are directed to simplify the algorithm structure and expand the application scope. Although the accuracy and convergence speed of QABA can be significantly improved by using multiple mechanisms in conjunction with each other, there are problems of lengthy structure and time-consuming operations. We will continue to find more efficient and concise evolutionary strategies to improve the performance of QABA and apply it to more complex combinatorial optimization problems.

## Figures and Tables

**Figure 1 sensors-23-00782-f001:**
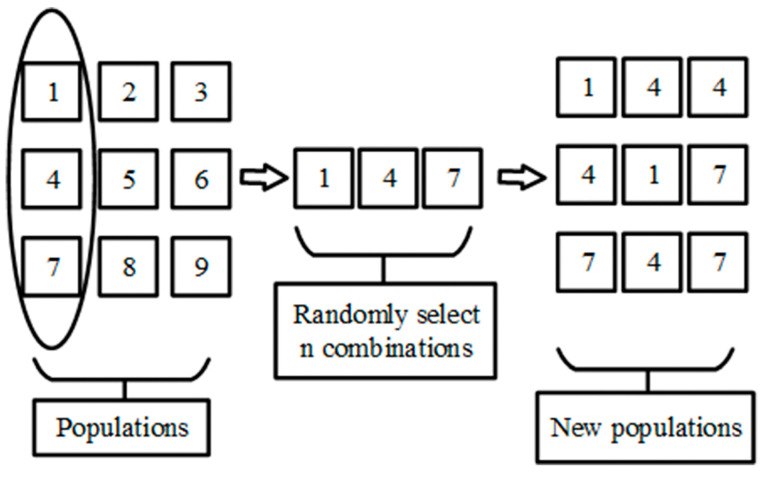
Tournament selection mechanism.

**Figure 2 sensors-23-00782-f002:**
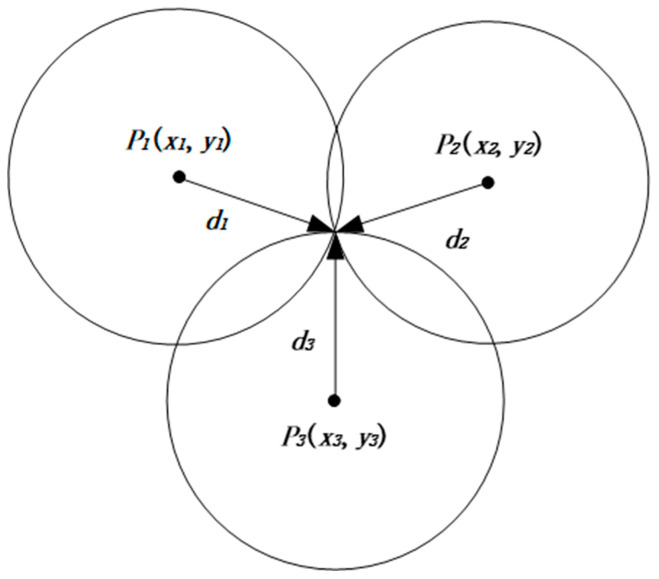
Trilateral positioning principle.

**Figure 3 sensors-23-00782-f003:**
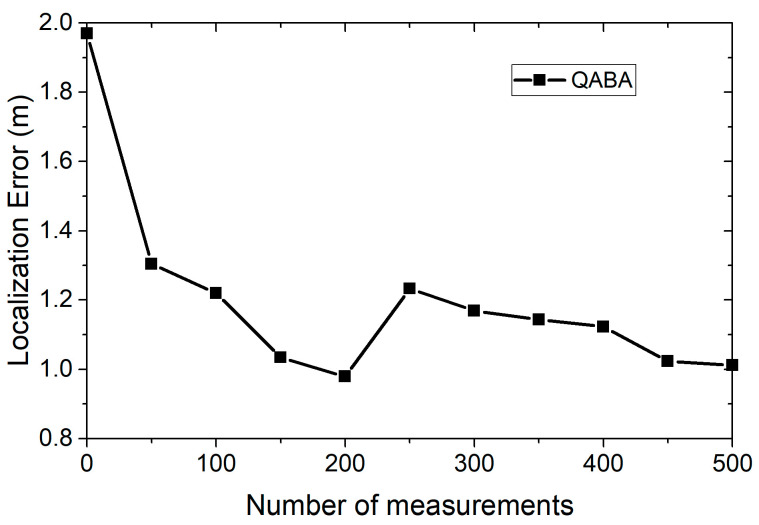
Localization error for different M values.

**Figure 4 sensors-23-00782-f004:**
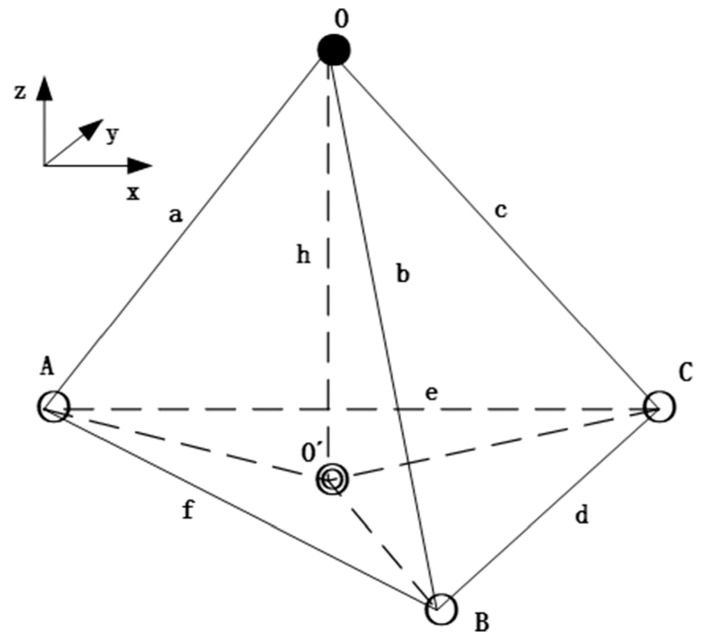
Geometric feature positioning principle.

**Figure 5 sensors-23-00782-f005:**
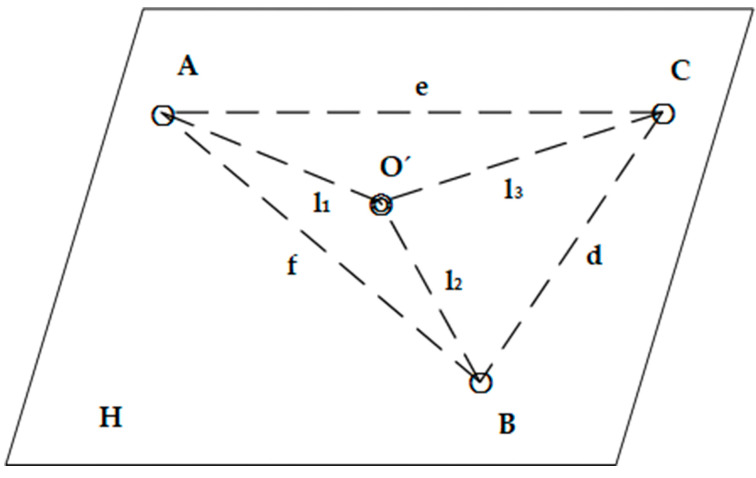
Geometric feature positioning principle.

**Figure 6 sensors-23-00782-f006:**
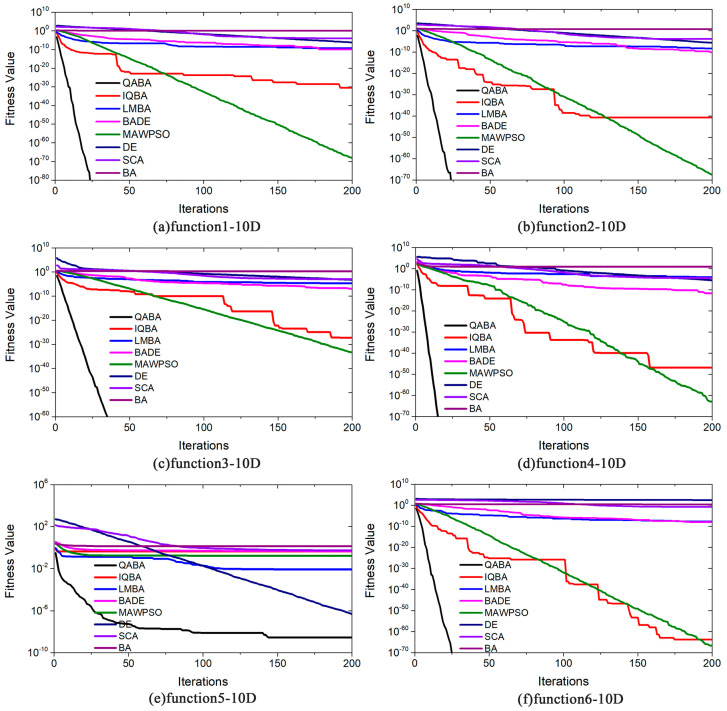
Comparison of optimization performance of different algorithms on functions (F1-6). (**a**) F1-10D; (**b**) F2-10D; (**c**) F3-10D; (**d**) F4-10D; (**e**) F5-10D; (**f**) F6-10D. The values of the Y-axis are shown in the curve as log10, but the curve still satisfies the rule of linear variation.

**Figure 7 sensors-23-00782-f007:**
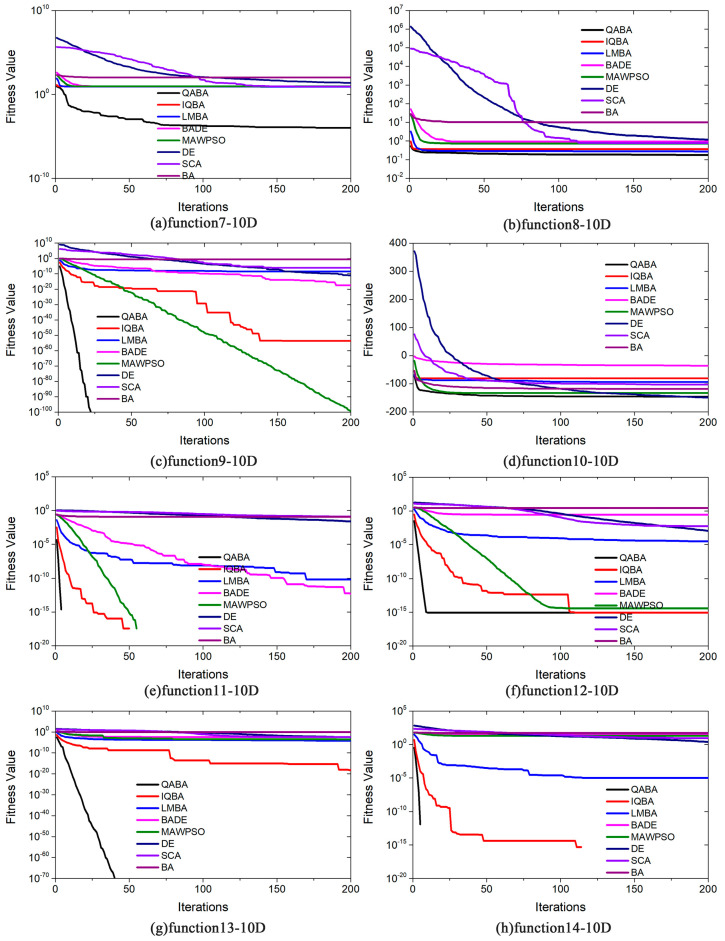
Comparison of optimization performance of different algorithms on functions (F7-14). (**a**) F7-10D; (**b**) F8-10D; (**c**) F9-10D; (**d**) F10-10D; (**e**) F11-10D; (**f**) F12-10D; (**g**) F13-10D; (**h**) F14-10D. In the curve, the y-value of the function F10−10D does not change, but the values of the other functions on the Y-axis are shown as log10, and the curve still satisfies the rule of linear variation.

**Figure 8 sensors-23-00782-f008:**
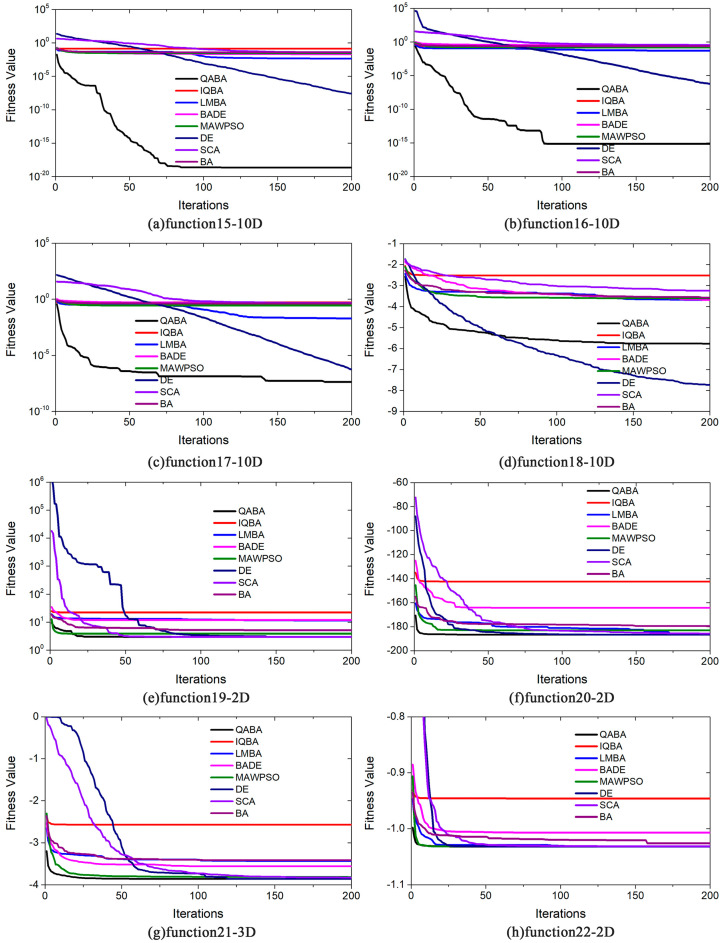
Comparison of optimization performance of different algorithms on functions (F15-22). (**a**) F15-10D; (**b**) F16-10D; (**c**) F17-10D; (**d**) F18-10D; (**e**) F19-2D; (**f**) F20-2D; (**g**) F21-3D; (**h**) F22-2D. In the curves, the y–values of functions F18-10D, F20-2D, F21-3D to 22-2D do not change, but the values of the other functions on the Y-axis are shown as log10, and the curves still satisfy the rule of linear variation.

**Figure 9 sensors-23-00782-f009:**
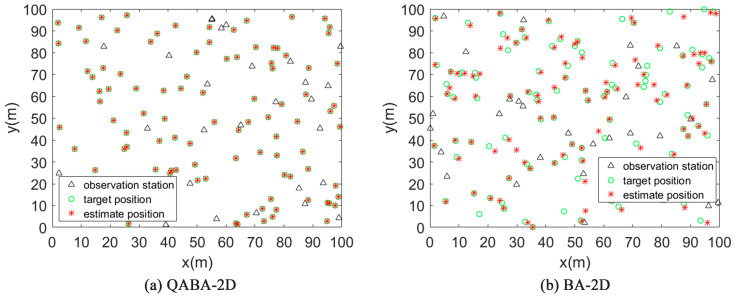
Comparison of positioning accuracy of QABA-2D and BA-2D. It can be seen from the figure that the positioning success rate and accuracy of QABA-2D are higher than those of BA-2D.

**Figure 10 sensors-23-00782-f010:**
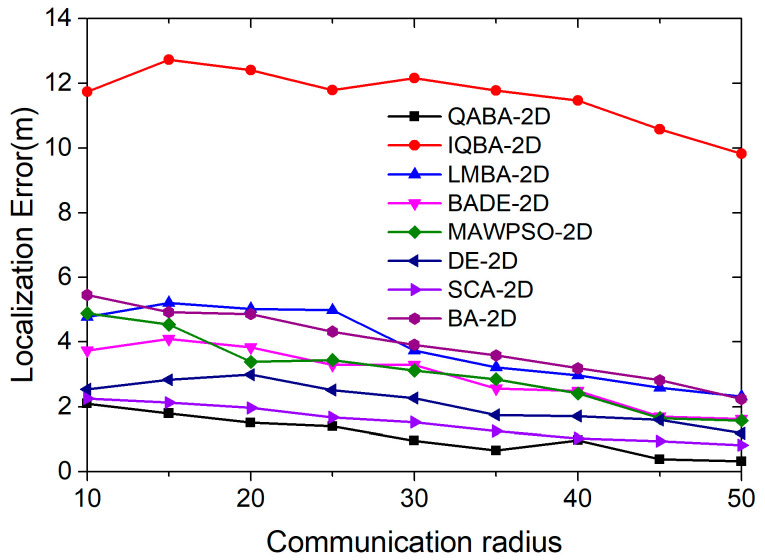
Relationship between communication radius and positioning error.

**Figure 11 sensors-23-00782-f011:**
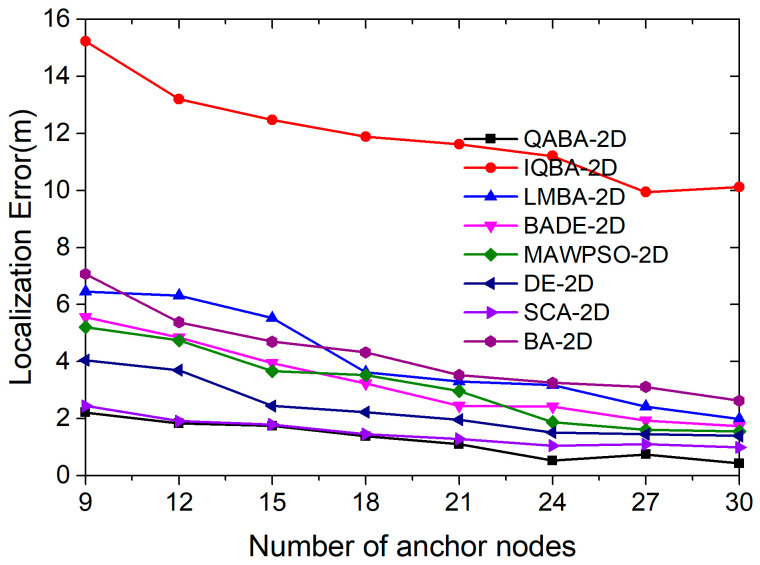
Relationship between the number of ANs and positioning error.

**Figure 12 sensors-23-00782-f012:**
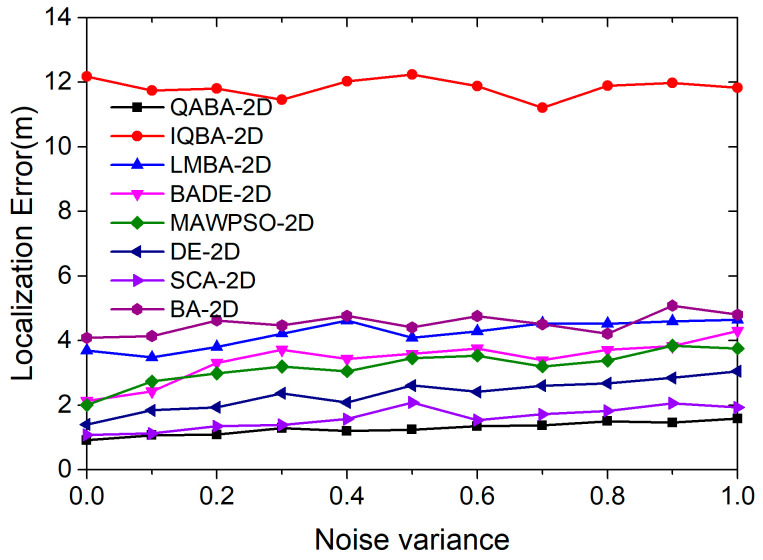
The relationship between different levels of noise and positioning error.

**Figure 13 sensors-23-00782-f013:**
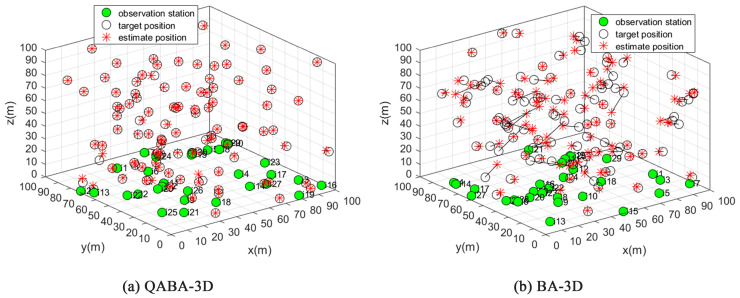
Comparison of QABA-3D and BA-3D positioning accuracy. It can be seen from the figure that the positioning success rate and accuracy of QABA-3D are higher than those of BA-3D.

**Figure 14 sensors-23-00782-f014:**
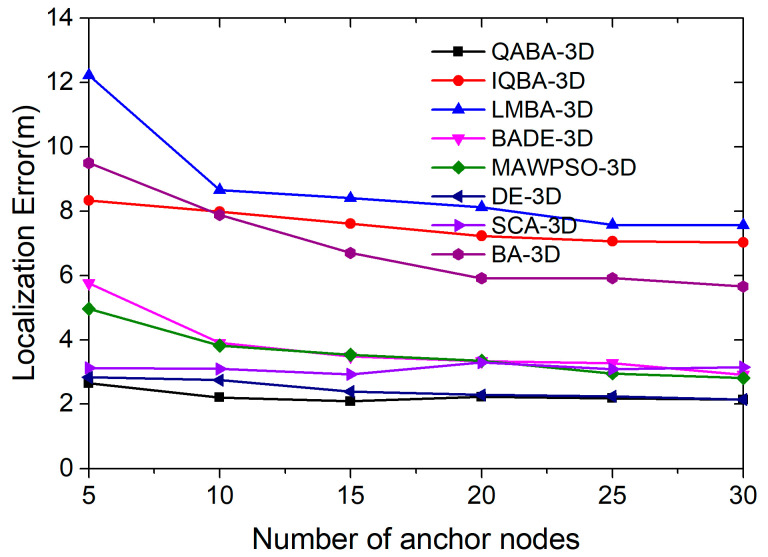
Relationship between the number of ANs and positioning error in 3D space.

**Figure 15 sensors-23-00782-f015:**
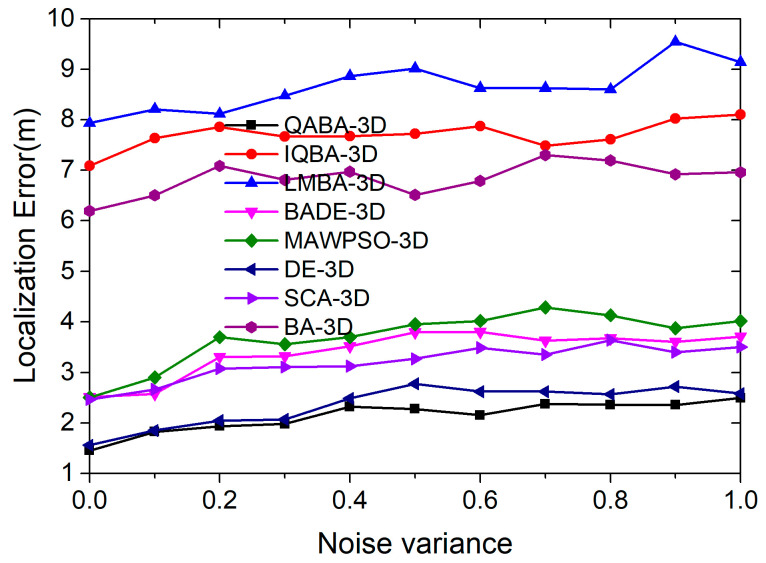
The relationship between noise variance and positioning error in 3D space.

**Table 1 sensors-23-00782-t001:** Parameters for 22 benchmarking functions.

No.	Function	Dim.	Parameter Range	Optimum
F1	Sphere Function	10	[−100,100]	0
F2	Sumsquares Function	10	[−10,10]	0
F3	Schwefel’s Problem 2.22	10	[−10,10]	0
F4	Table Function	10	[−100,100]	0
F5	Step Function	10	[−100,100]	0
F6	Zakharov Function	10	[−5,10]	0
F7	Rosenbrock Function	10	[−5,10]	0
F8	Dixon-Price Function	10	[−10,10]	0
F9	Sum of Different Powers Function	10	[−1,1]	0
F10	Trid Function	10	[−100,100]	−210
F11	Griewank Function	10	[−600,600]	0
F12	Ackley Function	10	[−30,30]	0
F13	Alpine Function	10	[−10,10]	0
F14	Rastrigin Function	10	[−5.12,5.12]	0
F15	Penalized 1 Function	10	[−50,50]	0
F16	Penalized 2 Function	10	[−50,50]	0
F17	Levy Function	10	[−10,10]	0
F18	Michalewicz Function	10	[0,π]	−9.6602
F19	Goldstein-Price Function	2	[−2,2]	3
F20	Shubert Function	2	[−10,10]	−186.7309
F21	Hartmann 3-D Function	3	[0,10]	−3.8628
F22	Six-Hump Camel Function	2	[−3,3], [−2,2]	−1.0316

**Table 2 sensors-23-00782-t002:** Comparison of the optimized values of different algorithms run 30 times.

No.	QABA (OV)	IQBA (OV)	LMBA (OV)	BADE (OV)	MAWPSO (OV)	DE (OV)	SCA (OV)	BA (OV)
F1	0.000	2.658 × 10^−137^(−)	7.060 × 10^−29^(−)	4.777 × 10^−35^(−)	2.475 × 10^−74^(−)	1.024 × 10^−7^(−)	1.411 × 10^−9^(−)	2.912 × 10^−1^(−)
F2	0.000	9.882 × 10^−148^(−)	8.600 × 10^−29^(−)	2.089 × 10^−38^(−)	6.502 × 10^−73^(−)	5.326 × 10^−7^(−)	7.317 × 10^−8^(−)	1.617 × 10^0^(−)
F3	0.000	1.843 × 10^−78^(−)	1.743 × 10^−15^(−)	4.767 × 10^−21^(−)	1.407 × 10^−36^(−)	3.070 × 10^−4^(−)	8.113 × 10^−5^(−)	1.116 × 10^0^(−)
F4	0.000	5.902 × 10^−145^(−)	1.203 × 10^−27^(−)	9.829 × 10^−37^(−)	2.553 × 10^−71^(−)	4.500 × 10^−7^(−)	2.961 × 10^−8^(−)	2.077 × 10^0^(−)
F5	2.982 × 10^−19^	3.813 × 10^−3^(−)	1.046 × 10^−6^(−)	3.920 × 10^−2^(−)	4.424 × 10^−2^(−)	1.150 × 10^−7^(−)	1.830 × 10^−1^(−)	4.111 × 10^−1^(−)
F6	0.000	2.767 × 10^−151^(−)	9.811 × 10^−29^(−)	5.401 × 10^−32^(−)	6.410 × 10^−74^(−)	1.078 × 10^2^(−)	2.010 × 10^−3^(−)	6.230 × 10^−1^(−)
F7	1.816 × 10^−6^	8.898 × 10^0^(−)	3.176 × 10^−1^(−)	8.716 × 10^0^(−)	8.650 × 10^0^(−)	3.748 × 10^0^(−)	7.494 × 10^0^(−)	2.914 × 10^1^(−)
F8	6.322 × 10^−2^	2.426 × 10^−1^(=)	2.263 × 10^−1^(=)	6.667 × 10^−1^(=)	6.733 × 10^−1^(=)	1.999 × 10^−1^(=)	6.668 × 10^−1^(=)	2.342 × 10^0^(=)
F9	0.000	6.107 × 10^−157^(−)	3.394 × 10^−21^(−)	2.748 × 10^−40^(−)	2.088 × 10^−106^(−)	8.898 × 10^−16^(−)	1.362 × 10^−18^(−)	1.142 × 10^−2^(−)
F10	−188.7238	−94.6994(−)	−113.6475(−)	−104.1850(−)	−185.4728(=)	−169.9349(=)	−121.8359(−)	−169.2080(=)
F11	0.000	0.000(=)	0.000(=)	0.000(=)	0.000(=)	1.057 × 10^−3^(−)	1.191 × 10^−6^(−)	3.485 × 10^−2^(−)
F12	8.882 × 10^−16^	8.882 × 10^−16^(=)	8.882 × 10^−16^(=)	8.882 × 10^−16^(=)	8.882 × 10^−16^(=)	4.984 × 10^−4^(−)	1.577 × 10^−4^(−)	1.835 × 10^0^(−)
F13	0.000	5.506 × 10^−70^(−)	7.231 × 10^−14^(−)	1.271 × 10^−19^(−)	1.571 × 10^−37^(−)	1.382 × 10^−3^(−)	6.960 × 10^−6^(−)	2.632 × 10^−1^(−)
F14	0.000	0.000(=)	0.000(=)	3.330 × 10^0^(−)	0.000(=)	7.544 × 10^−1^(−)	1.006 × 10^−4^(−)	2.700 × 10^1^(−)
F15	1.571 × 10^−32^	1.935 × 10^−2^(−)	1.010 × 10^−6^(−)	5.611 × 10^−4^(−)	3.327 × 10^−3^(−)	4.763 × 10^−9^(−)	1.642 × 10^−2^(−)	5.778 × 10^−3^(−)
F16	1.350 × 10^−32^	3.403 × 10^−4^(−)	1.183 × 10^−6^(−)	1.938 × 10^−2^(−)	2.381 × 10^−2^(−)	5.039 × 10^−8^(−)	1.500 × 10^−1^(−)	5.929 × 10^−2^(−)
F17	2.004 × 10^−23^	5.680 × 10^−3^(−)	8.969 × 10^−10^(−)	1.089 × 10^−1^(−)	9.429 × 10^−2^(−)	1.436 × 10^−7^(−)	2.350 × 10^−1^(−)	1.466 × 10^−1^(−)
F18	−7.7682	−5.7369(=)	−4.4415(=)	−5.5666(=)	−5.3906(=)	−8.3623(=)	−3.6644(=)	−5.2025(=)
F19	3.0000	3.0009(=)	3.0000(=)	3.0000(=)	3.0000(=)	3.0000(=)	3.0000(=)	3.0034(=)
F20	−186.7309	−186.3490(=)	−186.7309(=)	−186.7309(=)	−186.7309(=)	−186.7309(=)	−186.7109(=)	−186.7272(=)
F21	−3.8627	−3.8039(=)	−3.8540(=)	−3.8628(=)	−3.8627(=)	−3.8628(=)	−3.8615(=)	−3.8566(=)
F22	−1.0316	−1.0136(=)	−1.0316(=)	−1.0316(=)	−1.0316(=)	−1.0316(=)	−1.0316(=)	−1.0316(=)

**Table 3 sensors-23-00782-t003:** Comparison of mean and standard deviation results of different algorithms run 30 times.

No.	QABA (AV/SD)	IQBA (AV/SD)	LMBA (AV/SD)	BADE (AV/SD)	MAWPSO (AV/SD)	DE (AV/SD)	SCA (AV/SD)	BA (AV/SD)
F1	0.000/0.000	2.743 × 10^−31^/1.480 × 10^−30^(−)	4.584 × 10^−10^/1.570 × 10^−9^(−)	8.720 × 10^−11^/4.154 × 10^−10^(−)	6.714 × 10^−69^/1.927 × 10^−68^(−)	5.711 × 10^−7^/4.067 × 10^−7^(−)	9.423 × 10^−5^/3.783 × 10^−4^(−)	9.552 × 10^−1^/4.532 × 10^−1^(−)
F2	0.000/0.000	1.977 × 10^−41^/1.060 × 10^−40^(−)	4.704 × 10^−9^/1.950 × 10^−8^(−)	7.812 × 10^−11^/2.978 × 10^−10^(−)	2.796 × 10^−68^/4.605 × 10^−68^(−)	2.477 × 10^−6^/1.360 × 10^−6^(−)	1.198 × 10^−4^/2.805 × 10^−4^(−)	5.836 × 10^0^/2.121 × 10^0^(−)
F3	1.301 × 10^−309^/0.000	5.869 × 10^−28^/3.160 × 10^−27^(−)	2.406 × 10^−5^/5.680 × 10^−5^(−)	1.226 × 10^−7^/3.977 × 10^−7^(−)	5.030 × 10^−34^/1.205 × 10^−33^(−)	6.179 × 10^−4^/2.489 × 10^−4^(−)	1.136 × 10^−3^/1.742 × 10^−3^(−)	2.393 × 10^0^/5.264 × 10^−1^(−)
F4	0.000/0.000	1.576 × 10^−47^/8.490 × 10^−47^(−)	9.507 × 10^−5^/2.190 × 10^−4^(−)	2.507 × 10^−12^/1.067 × 10^−11^(−)	1.489 × 10^−63^/8.017 × 10^−63^(−)	3.517 × 10^−6^/2.347 × 10^−6^(−)	3.506 × 10^−5^/6.369 × 10^−5^(−)	8.168 × 10^0^/5.222 × 10^0^(−)
F5	2.819 × 10^−9^/4.534 × 10^−9^	4.323 × 10^−1^/4.601 × 10^−1^(−)	8.421 × 10^−3^/1.396 × 10^−2^(−)	4.551 × 10^−1^/2.892 × 10^−1^(−)	1.661 × 10^−1^/1.174 × 10^−1^(−)	4.880 × 10^−7^/2.534 × 10^−7^(−)	5.476 × 10^−1^/1.866 × 10^−1^(−)	1.340 × 10^0^/6.187 × 10^−1^(−)
F6	0.000/0.000	1.533 × 10^−64^/8.250 × 10^−64^(−)	1.785 × 10^−8^/5.157 × 10^−8^(−)	7.955 × 10^−9^/3.032 × 10^−8^(−)	1.976 × 10^−67^/1.010 × 10^−66^(−)	2.589 × 10^2^/7.898 × 10^1^(−)	1.395 × 10^−1^/2.224 × 10^−1^(−)	2.302 × 10^0^/1.370 × 10^0^(−)
F7	1.026 × 10^−4^/1.957 × 10^−4^	8.911 × 10^0^/5.513 × 10^−3^(−)	8.131 × 10^0^/2.113 × 10^0^(−)	8.927 × 10^0^/4.513 × 10^−2^(−)	8.880 × 10^0^/8.114 × 10^−2^(−)	2.323 × 10^1^/1.317 × 10^1^(−)	8.658 × 10^0^/1.636 × 10^0^(−)	1.008 × 10^2^/4.561 × 10^1^(−)
F8	1.791 × 10^−1^/5.924 × 10^−2^	3.626 × 10^−1^/1.604 × 10^−1^(=)	2.748 × 10^−1^/4.146 × 10^−2^(=)	9.245 × 10^−1^/9.194 × 10^−2^(=)	7.256 × 10^−1^/5.103 × 10^−2^(=)	1.179 × 10^0^/1.861 × 10^0^(=)	7.717 × 10^−1^/3.806 × 10^−1^(=)	9.981 × 10^0^/6.783 × 10^0^(=)
F9	0.000/0.000	1.992 × 10^−54^/1.070 × 10^−53^(−)	2.817 × 10^−9^/1.172 × 10^−8^(−)	3.252 × 10^−18^/1.746 × 10^−17^(−)	4.951 × 10^−100^/1.473 × 10^−99^(−)	8.163 × 10^−12^/1.818 × 10^−11^(−)	7.740 × 10^−7^/4.136 × 10^−6^(−)	2.295 × 10^−1^/2.039 × 10^−1^(−)
F10	−146.5176/1.435 × 10^1^	−80.9237/8.899 × 10^0^(−)	−94.1017/5.741 × 10^0^(−)	−36.3750/2.365 × 10^1^(−)	−133.8308/3.389 × 10^1^(−)	−149.5293/1.344 × 10^1^(=)	−103.4097/9.065 × 10^0^(−)	−118.8001/2.131 × 10^1^(−)
F11	0.000/0.000	0.000/0.000(=)	6.655 × 10^−11^/1.719 × 10^−10^(−)	5.361 × 10^−13^/2.592 × 10^−12^(−)	0.000/0.000(=)	2.484 × 10^−2^/1.632 × 10^−2^(−)	1.328 × 10^−1^/1.347 × 10^−1^(−)	1.207 × 10^−1^/4.767 × 10^−2^(−)
F12	8.882 × 10^−16^/0.000	8.882 × 10^−16^/0.000(=)	3.260 × 10^−5^/9.776 × 10^−5^(−)	2.810 × 10^−1^/7.249 × 10^−1^(−)	3.849 × 10^−15^/1.324 × 10^−15^(=)	1.096 × 10^−3^/3.228 × 10^−4^(−)	5.621 × 10^−3^/1.062 × 10^−2^(−)	2.727 × 10^0^/4.061 × 10^−1^(−)
F13	1.785 × 10^−306^/0.000	6.224 × 10^−19^/2.650 × 10^−18^(−)	5.990 × 10^−5^/2.306 × 10^−4^(−)	3.797 × 10^−3^/4.400 × 10^−3^(−)	2.495 × 10^−4^/1.343 × 10^−3^(−)	3.305 × 10^−3^/1.167 × 10^−3^(−)	3.449 × 10^−3^/3.368 × 10^−3^(−)	9.019 × 10^−1^/4.235 × 10^−1^(−)
F14	0.000/0.000	0.000/0.000(=)	1.047 × 10^−5^/5.512 × 10^−5^(−)	3.051 × 10^1^/1.320 × 10^1^(−)	2.003 × 10^1^/1.089 × 10^1^(−)	2.381 × 10^0^/1.000 × 10^0^(−)	8.960 × 10^0^/1.191 × 10^1^(−)	5.209 × 10^1^/1.025 × 10^1^(−)
F15	2.269 × 10^−19^/1.220 × 10^−18^	1.384 × 10^−1^/8.445 × 10^−2^(−)	4.299 × 10^−3^/5.766 × 10^−3^(−)	1.935 × 10^−2^/2.142 × 10^−2^(−)	2.848 × 10^−2^/2.751 × 10^−2^(−)	2.525 × 10^−8^/2.233 × 10^−8^(−)	4.023 × 10^−2^/2.016 × 10^−2^(−)	3.643 × 10^−2^/2.248 × 10^−2^(−)
F16	7.400 × 10^−16^/3.980 × 10^−15^	2.712 × 10^−1^/2.117 × 10^−1^(−)	5.277 × 10^−2^/7.577 × 10^−2^(−)	3.315 × 10^−1^/7.577 × 10^−2^(−)	1.620 × 10^−1^/1.078 × 10^−1^(−)	5.761 × 10^−7^/8.659 × 10^−7^(−)	3.632 × 10^−1^/9.188 × 10^−2^(−)	2.373 × 10^−1^/8.140 × 10^−2^(−)
F17	4.284 × 10^−8^/1.030 × 10^−7^	5.167 × 10^−1^/3.457 × 10^−1^(−)	1.985 × 10^−2^/2.812 × 10^−2^(−)	4.691 × 10^−1^/1.901 × 10^−1^(−)	2.793 × 10^−1^/9.907 × 10^−2^(−)	5.615 × 10^−7^/3.630 × 10^−7^(−)	4.730 × 10^−1^/1.124 × 10^−1^(−)	4.286 × 10^−1^/1.629 × 10^−1^(−)
F18	−5.7820/1.101 × 10^0^	−2.5298/7.607 × 10^−1^(=)	−3.6944/5.159 × 10^−1^(=)	−3.6952/9.096 × 10^−1^(=)	−3.6069/7.455 × 10^−1^(=)	−7.7391/2.998 × 10^−1^(=)	−3.2484/2.458 × 10^−1^(=)	−3.5756/6.495 × 10^−1^(=)
F19	3.0093/3.135 × 10^−2^	22.4502/2.025 × 10^1^(−)	11.6056/1.236 × 10^1^(−)	12.0056/2.124 × 10^1^(−)	3.9032/4.846 × 10^0^(=)	3.0034/1.829 × 10^−2^(=)	3.0011/2.397 × 10^−3^(=)	5.2367/6.072 × 10^0^(=)
F20	−186.7307/5.618 × 10^−4^	−142.4349/4.594 × 10^1^(−)	−186.2162/9.798 × 10^−1^(=)	−164.2457/3.054 × 10^1^(=)	−183.0969/1.925 × 10^1^(=)	−186.7268/7.362 × 10^−3^(=)	−185.7394/1.058 × 10^0^(=)	−179.5836/1.088 × 10^1^(=)
F21	−3.8615/1.097 × 10^−3^	−2.5718/8.506 × 10^−1^(=)	−3.4375/3.661 × 10^−1^(=)	−3.5619/4.712 × 10^−1^(=)	−3.8166/7.504 × 10^−2^(=)	−3.8628/5.099 × 10^−10^(=)	−3.8284/2.767 × 10^−2^(=)	−3.4144/4.684 × 10^−1^(=)
F22	−1.0316/1.970 × 10^−7^	−0.9460/1.001 × 10^−1^(=)	−1.0316/1.384 × 10^−4^(=)	−1.0068/9.946 × 10^−2^(=)	−1.0316/5.019 × 10^−5^(=)	−1.0316/6.906 × 10^−15^(=)	−1.0315/1.411 × 10^−4^(=)	−1.0260/1.215 × 10^−2^(=)

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
