# Peer review of "A Quantum Annealing Bat Algorithm for Node Localization in Wireless Sensor Networks"

_sensors, 2023, doi:10.3390/s23020782_

Round 1
Reviewer 1 Report
In this work, the authors presented a quantum annealing Bat algorithm for node localization in wireless sensor networks. Some of the suggestions are listed below:
1. There are many numerical equations that lack proper explanation, especially equations 18-21, and so on. Please explain every equation properly.
2. There are many grammatical mistakes and typos that must be corrected with detailed proofreading. Please revise the manuscript carefully to avoid such errors.
3. In figure 11, the X-axis can be X(m) and y-axis can be Y(m). Please modify.
4. Most of the figures are very, especially figure 15. Please add clear figures.
5. The author’s work has very limited novelty because there are several approaches available dealing with node localization in wireless sensor networks. What is new in this work?
6. Why have the authors repeated the localization for several iterations? Why I can’t keep up to 1 iteration?
7. What is the benefit of many iterations?
8. I can’t see any major difference between figures 6-10, except the number of iterations.
9. Can this model be implemented in an underwater environment as well? If not, then why? And if yes, then how?
10. Reference 63-68 seems irrelevant and belongs to similar authors; they should be removed. Please revise the remaining references as well and remove all irrelevant references.
Author Response
On behalf of my co-authors, we appreciate the editor and the reviewer very much for the positive and constructive comments and suggestions on our manuscript.
Manuscript ID: sensors-2136139,
Title: A Quantum Annealing Bat Algorithm for Node Localization in Wireless Sensor Networks.
We carefully studied the comments and found the corresponding questions in the manuscript. We have tried our best to revise our manuscript according to the comments. Attached is a revised version that we would like to submit for your consideration. We would like to express our great appreciation to the editor and reviewers for comments on our paper. Looking forward to hearing from you.

Reviewer 2 Report
The authors propose the employment of the QABA to tackle high-complexity optimization problems. Later, they use this algorithm for the node location problem in wireless sensor networks, attaining promising results and analyzing different situations, proving their method's benefits in authentic contexts. The manuscript is well-structured and presents the results in a fashionable manner.
Although the authors present an interesting work, there are still some concerns to be addressed to improve the overall quality of the research:
1- The authors misunderstood the terms optimal and optimized along the text. When we are talking about metaheuristics in which the achievement of the optimal solution is not guaranteed, the term optimized is more correct. Please revise accordingly.
2- Related works are only referring to general metaheuristics but the achievement of optimized solutions does require the employment of local search procedures to refine the results. Since this is a key point in your proposal, I would recommend a brief discussion on the hybridization of local search with the metaheuristics in the field including papers such as:
Díez-González, J.; Verde, P.; Ferrero-Guillén, R.; Álvarez, R.; Pérez, H. Hybrid Memetic Algorithm for the Node Location Problem in Local Positioning Systems. Sensors 2020, 20, 5475. https://doi.org/10.3390/s20195475
Manjarres, D., Del Ser, J., Gil-Lopez, S., Vecchio, M., Landa-Torres, I., & Lopez-Valcarce, R. (2013). A novel heuristic approach for distance-and connectivity-based multihop node localization in wireless sensor networks. Soft Computing, 17(1), 17-28.
3- The number of ANs is critical to achieving improved results in localization. In this work seems that the authors have used a constant number of nodes to calculate the location. How have they made this position calculation? How many nodes have they employed? How do they select the nodes?
4- How do the authors explain the increase in the localization error with a higher number of measurements in Figure 3?
5- Please clarify within the text the differences of this research with Referece [55].
6- How do the authors consider noise during the experiments? This should be done in a heteroscedastic way such as reference:
Huang, B.; Xie, L.; Yang, Z. TDOA-based source localization with distance-dependent noises. IEEE Trans. Wirel. Commun. 2014, 14, 468–480.
7- I suggest including the best result over the 30 runs of each algorithm in Table 2.
8- The effect of the communication radius and the number of ANs in the positioning error depends on an optimal selection of the anchors used to determine the target location as demonstrated in:
Álvarez, R., Díez-González, J., Verde, P., Ferrero-Guillén, R., & Perez, H. (2023). Combined sensor selection and node location optimization for reducing the localization uncertainties in wireless sensor networks. Ad Hoc Networks, 139, 103036.
Please clarify this within the text while discussing.
I strongly believe that these proposed changes can help improve the quality of the paper and enhance the discussion of the results. Then, the paper could be publish since it is a well-conducted work.
Author Response

(The authors gave the same response as above.)

Round 2
Reviewer 1 Report
No more comments. My comments are addressed well and the manuscript is improved.
Reviewer 2 Report
The authors have made considerable effort in the revised version of the manuscript. I consider that they have addressed all my concerns. Therefore, the paper can be published.